# Prophylactic versus therapeutic dose anticoagulation effects on survival among critically ill patients with COVID-19

**Wouter S. Hoogenboom**[1‡]*, **Joyce Q. Lu**[1‡], **Benjamin Musheyev**[2], **Lara Borg**[2], **Rebeca Janowicz**[2], **Stacey Pamlayne**[2], **Wei Hou**[2], **Tim Q. Duong**[1]*

**1** Albert Einstein College of Medicine and Montefiore Medical Center, Bronx, New York, United States of America, **2** Stony Brook University, Renaissance School of Medicine, Stony Brook, New York, United States of America

‡ WSH and JQL are co-first authors on this work.
* wouter.hoogenboom@einsteinmed.org (WSH); tim.duong@einsteinmed.org (TQD)

**Data Availability Statement:** Data are available at Kaggle: https://www.kaggle.com/joycelu9087/anticoagulant-data.

## Abstract

### Introduction

Although patients with severe COVID-19 are known to be at high risk of developing thrombotic events, the effects of anticoagulation (AC) dose and duration on in-hospital mortality in critically ill patients remain poorly understood and controversial. The goal of this study was to investigate survival of critically ill COVID-19 patients who received prophylactic or therapeutic dose AC and analyze the mortality rate with respect to detailed demographic and clinical characteristics.

### Materials and methods

We conducted a retrospective, observational study of critically ill COVID-19 patients admitted to the ICU at Stony Brook University Hospital in New York who received either prophylactic (n = 158) or therapeutic dose AC (n = 153). Primary outcome was in-hospital death assessed by survival analysis and covariate-adjusted Cox proportional hazard model.

### Results

For the first 3 weeks of ICU stay, we observed similar survival curves for prophylactic and therapeutic AC groups. However, after 3 or more weeks of ICU stay, the therapeutic AC group, characterized by high incidence of acute kidney injury (AKI), had markedly higher death incidence rates with 8.6 deaths (95% CI = 6.2–11.9 deaths) per 1,000 person-days and about 5 times higher risk of death (adj. HR = 4.89, 95% CI = 1.71–14.0, p = 0.003) than the prophylactic group (2.4 deaths [95% CI = 0.9–6.3 deaths] per 1,000 person-days). Among therapeutic AC users with prolonged ICU admission, non-survivors were characterized by older males with depressed lymphocyte counts and cardiovascular disease.

**Funding:** The author(s) received no specific funding for this work.

**Competing interests:** The authors have declared that no competing interests exist.

## Conclusions

Our findings raise the possibility that prolonged use of high dose AC, independent of thrombotic events or clinical background, might be associated with higher risk of in-hospital mortality. Moreover, AKI, age, lymphocyte count, and cardiovascular disease may represent important risk factors that could help identify at-risk patients who require long-term hospitalization with therapeutic dose AC treatment.

## Introduction

Emerging evidence indicates that patients with severe COVID-19, caused by severe acute respiratory syndrome coronavirus 2 (SARS-CoV-2) [1], are at increased risk of developing thrombotic events [2, 3]. Abnormal coagulation parameters are commonly observed in severe COVID-19 patients and associated with thrombotic complications and high mortality [2, 4]. High incidence rates of thrombotic events up to 69% have been observed in COVID-19 patients in the ICU [5–7], markedly higher than non-COVID-19 patients with acute respiratory distress syndrome (ARDS) [2]. Autopsy studies on patients who died from COVID-19 report widespread microscopic thrombosis as one of the main causes of death [8, 9]. To mitigate vascular complications and improve patient outcomes, clinical guidelines and consensus documents recommend the use of anticoagulants (AC) in all patients hospitalized for COVID-19 [10, 11].

However, the survival benefit and dosage level of anticoagulation remains controversial and understudied often with relatively small sample size and limited clinical variables. One study reported no significant AC dosing effect on 28-day survival [12], two studies reported lower mortality associated with high dose AC compared to medium or low dose AC [13, 14], and one study reported increased mortality and more adverse events associated with therapeutic dose AC [15]. In a multicenter randomized trial [16], intermediate dose prophylactic anticoagulation did not result in a significant difference in venous or arterial thrombosis, treatment with extracorporeal membrane oxygenation, or 30-day mortality as compared to standard-dose prophylactic anticoagulation. Other studies that examined the effects of anticoagulants on COVID-19 outcomes did not specifically focus on treatment dose level or ICU populations [17–23].

While more data is needed to assess the long-term effects of AC treatment on COVID-19 outcomes, most COVID-19 studies are limited to reporting acute effects (3 to 4 weeks survival). Approximately one quarter of COVID-19 patients who receive invasive mechanical ventilation require ventilation support for more than 4 weeks [24], but there is limited data on the effects of prolonged AC use during hospitalization among these patients. The goal of this study is to investigate further survival of critically ill COVID-19 patients who received prophylactic or therapeutic AC dose and analyze the in-hospital mortality rate with respect to detailed demographic and clinical characteristics. We hypothesize that prolonged treatment with therapeutic dose AC provides no survival benefit over prophylactic dose AC for severe COVID-19 patients admitted to the ICU.

## Materials and methods

### Study population and data collection

This retrospective, single-center study from Stony Brook University Hospital was approved by the Stony Brook University Institutional Review Board with an exemption for informed

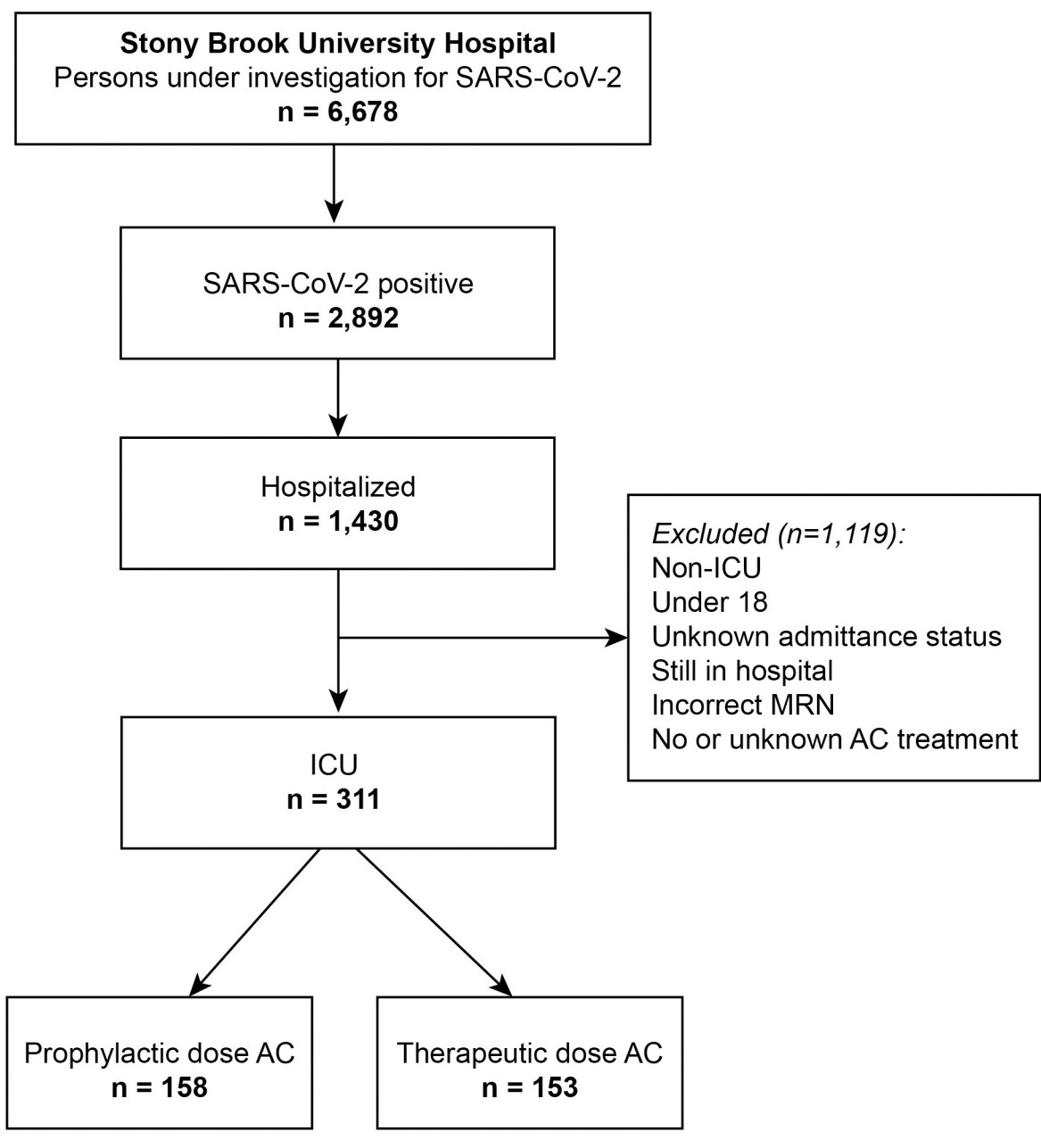

**Fig 1. Patient selection flowchart.**

consent (IRB#: IRB2020-00207). A flowchart of patient selection is presented in Fig 1. Data were obtained from the COVID-19 Persons Under Investigation (PUI) registry (n = 6,678) of the Stony Brook University Hospital emergency department from February 7, 2020, to June 30, 2020. Of the 2,892 individuals who tested positive for SARS-CoV-2 infection by real-time polymerase chain reaction test on a nasopharyngeal swab specimen, 1430 were hospitalized. Patients not admitted or upgraded to the ICU, under 18 years old, still in the hospital at the time of this analysis, not primarily admitted for COVID-19 diagnosis (e.g., trauma), with no or unknown AC treatment, with incorrect MRN, or placed on comfort care early on without

escalated care (ICU) were excluded. After we applied these exclusion criteria, a total of 311 patients with COVID-19 admitted to the ICU comprised the final sample size of our study.

Treatment was part of routine clinical care at Stony Brook University Hospital. Patients that did not receive any anticoagulation had contraindications such as high risk for bleeding or supratherapeutic INR at admission. Dose regimens were based on general risk assessment considering clinical background, preexisting conditions, and presenting laboratory values.

Patients were divided into one of two groups based on AC treatment dosage received: (1) continuous and exclusively low dose / prophylactic anticoagulation—that is, subcutaneous heparin or Lovenox (enoxaparin) at a dose of 40 mg twice daily; or (2) high dose / therapeutic anticoagulation—that is a) any heparin drip; b) Eliquis (apixaban), Xarelto (rivaroxaban), Pradaxa (dabigatran), or Coumadin (warfarin) at a typical therapeutic dose strength; or c) Lovenox (enoxaparin) at a dose of 1 mg/kg twice daily or 1.5 mg/kg daily. Patients who received both prophylactic and therapeutic dosages during their hospital stay were grouped in the therapeutic cohort for this analysis. Patients with contraindications for anticoagulant treatment who did not receive anticoagulation or who required a pause in anticoagulation during hospital stay were excluded from the analysis.

We collected COVID-19 outcome variables that included mortality rate, ICU duration, oxygen therapy, invasive mechanical ventilator (IMV) use, and incidence of acute kidney injury (AKI). We also collected demographics (i.e., age, gender, ethnicity and race), chronic comorbidities (i.e., smoking history, diabetes, hypertension, asthma, chronic obstructive pulmonary disease (COPD), coronary artery disease (CAD), heart failure, cancer, immune- suppression, chronic kidney disease (CKD)), vitals (i.e., heart rate (HR), diastolic blood pressure (DBP), systolic blood pressure (SBP), respiratory rate, pulse oxygen saturation ($SpO_2$) and temperature), coagulation parameters (D-dimer, platelet count), and other laboratory tests (i.e., creatinine, procalcitonin, aspartate transaminase (AST), alanine aminotransferase (ALT), ferritin, lactate dehydrogenase (LDH), white blood cell count (WBC), C-reactive protein (CRP), lymphocytes count, brain natriuretic peptide (BNP), albumin, hematocrit, bicarbonate, creatinine), and blood gas variables (i.e., $pO_2$, $pCO_2$, and pH). All patient characteristics, symptoms, vitals, and laboratory values were collected at admission.

## Statistical analysis

All statistical analyses were performed using Stata statistical software (version 13.1, StataCorp, College Station, TX). Pairwise comparisons of continuous variables, reported in medians and interquartile ranges (IQR), were performed using Mann Whitney U tests. Categorical variables were compared using Fisher's exact test. The primary outcome was mortality. We constructed Kaplan-Meier survival plots and tested the equality of survival functions between prophylactic and therapeutic anticoagulated cohorts with the log-rank test. Risk of death was further assessed using Cox's proportional hazards model with Breslow method for ties and adjusted for group differences in age. The data are reported as hazard ratio (HR) with 95% confidence intervals (CIs) and incidence rates standardized to 1,000 person-days. Given multiple outcomes of interest, we considered a more stringent two-sided $\alpha$ of 0.01 to determine statistical significance.

## Results

Of the 311 COVID-19 patients admitted to the ICU, 158 received prophylactic dose AC, and 153 received therapeutic dose AC. Characteristics of critically ill COVID-19 patients stratified by AC dose are presented in Table 1.

**Table 1. Demographics, clinical variables and escalated care of critically ill COVID-19 patients who received prophylactic or therapeutic dose AC.**

| | Prophylactic dose | Therapeutic dose | p-value |
|---|---|---|---|
| | n = 158 | n = 153 | |
| **Demographics** | | | |
| Age in years, median (IQR) | 56 (48–67) | 63 (53–72) | 0.002 |
| Female sex, n (%) | 53 (33) | 43 (28) | 0.46 |
| Race, n (%) | | | 0.80 |
| White | 71 (44) | 70 (46) | |
| Black | 11 (7) | 9 (6) | |
| Asian | 8 (5) | 12 (8) | |
| American Indian | 1 (0.6) | 1 (0.7) | |
| Unknown | 69 (43) | 59 (39) | |
| Ethnicity, n (%) | | | 0.55 |
| Hispanic | 43 (27) | 43 (28) | |
| Non-Hispanic | 88 (55) | 87 (58) | |
| Unknown | 29 (18) | 21 (14) | |
| **COVID-19 symptoms**, n (%) | | | |
| Asymptomatic | 6 (4) | 6 (4) | >0.99 |
| Chest Discomfort, chest pain | 29 (18) | 19 (13) | 0.16 |
| Cough | 112 (70) | 99 (66) | 0.40 |
| Diarrhea | 32 (20) | 33 (22) | 0.78 |
| Fatigue | 30 (19) | 31 (21) | 0.67 |
| Fever (patient reported) | 114 (71) | 97 (64) | 0.28 |
| Headache | 22 (14) | 10 (7) | 0.04 |
| Loss of smell | 5 (3) | 8 (5) | 0.17 |
| Loss of taste | 6 (4) | 8 (5) | 0.28 |
| Myalgia | 42 (26) | 28 (19) | 0.17 |
| Nausea or vomiting | 28 (18) | 21 (14) | 0.35 |
| Runny nose/nasal congestion | 9 (6) | 7 (5) | 0.80 |
| Shortness of breath | 117 (73) | 111 (74) | >0.99 |
| Sore throat | 15 (9) | 8 (5) | 0.19 |
| Sputum | 15 (9) | 10 (7) | 0.41 |
| **Comorbidities**, n (%) | | | |
| Asthma | 14 (9) | 13 (9) | >0.99 |
| Cancer | 8 (5) | 8 (5) | >0.99 |
| Chronic kidney disease | 11 (7) | 14 (9) | 0.54 |
| COPD | 4 (3) | 16 (10) | 0.005 |
| Coronary artery disease | 17 (11) | 26 (17) | 0.14 |
| Heart failure | 10 (6) | 11 (7) | 0.82 |
| Hypertension | 66 (42) | 83 (54) | 0.03 |
| Immunosuppression | 6 (4) | 15 (10) | 0.04 |
| Type-2 Diabetes | 49 (31) | 43 (28) | 0.62 |
| Smoking History | | | 0.24 |
| Current Smoker | 5 (3) | 6 (4) | |
| Former Smoker | 27 (17) | 35 (23) | |
| Never Smoker | 118 (74) | 95 (63) | |
| Unknown | 10 (6) | 15 (10) | |
| **Vital signs**, median (IQR) | | | |
| Heart rate, bpm | 94 (81–104) | 95 (82–105) | 0.87 |

*(Continued)*

**Table 1.** (Continued)

| | Prophylactic dose | Therapeutic dose | p-value |
|---|---|---|---|
| | **n = 158** | **n = 153** | |
| Diastolic blood pressure, mmHg | 71 (66–78) | 69 (63–75) | 0.06 |
| Respiratory rate, rate/min | 23 (19–30) | 24 (20–30) | 0.65 |
| Oxygen saturation, % | 94 (92–96) | 94 (91–95) | 0.12 |
| Systolic blood pressure, mmHg | 121 (113–134) | 122 (111–135) | 0.83 |
| Temperature, ˚C | 37.3 (36.9–37.8) | 37.0 (36.7–37.4) | 0.004 |
| **Laboratory values**, median, (IQR) | | | |
| Alanine aminotransferase, U/L | 37 (24–66) | 38 (22–63) | 0.93 |
| Aspartate aminotransferase, U/L | 45 (33–77) | 49 (34–76) | 0.67 |
| Bicarbonate, mEg/L | 22 (19–24) | 22 (19–24) | 0.83 |
| BNP, pg/mL | 173 (62–624) | 433 (110–1477) | 0.002 |
| C-reactive protein, mg/dL | 14 (7–22) | 15 (7–26) | 037 |
| Creatinine mg/dL | 0.9 (0.7–1.1) | 1 (0.7–1.6) | 0.011 |
| Ferritin, ng/mL | 978 (520–1689) | 1185 (633–1922) | 0.24 |
| Hematocrit, % | 40 (35–43) | 40 (35–44) | 0.91 |
| Lactate dehydrogenase, U/L | 416 (321–589) | 527 (383–690) | <0.001 |
| Lymphocytes (lymp/mcL) | 11 (6–17) | 7 (4–11) | <0.001 |
| $paCO_2$, mmHg | 40 (34–46) | 39 (33–50) | 0.61 |
| $paO_2$, mmHg | 87 (70–134) | 76 (61–96) | <0.001 |
| pH | 7.4 (7.3–7.5) | 7.4 (7.3–7.5) | 0.64 |
| Procalcitonin, ng/mL | 0.3 (0.2–0.7) | 0.3 (0.2–0.8) | 0.29 |
| White blood cell count, $x10^3$ /ml | 8 (6–11) | 9 (6–15) | 0.02 |
| **AC parameters**, median, (IQR) | | | |
| D-dimer, ng/mL | 429 (275–798) | 659 (375–2043) | <0.001 |
| Platelet count, $10^9$/L | 211 (154–278) | 207 (161–277) | 0.76 |
| **Thrombotic events**, n (%) | | | |
| Pulmonary embolism | 3 (2) | 12 (8) | 0.02 |
| Deep vein thrombosis | 1 (0.7) | 11 (7) | 0.002 |
| Myocardial infarction | 10 (6) | 22 (15) | 0.02 |
| Stroke | 8 (5) | 13 (9) | 0.26 |
| Limb ischemia | 0 (0) | 3 (2) | 0.12 |
| Renal/mesenteric infarct | 1 (1) | 2 (1) | 0.62 |
| Other clotting events | 0 (0) | 2 (1) | 0.24 |
| **Clinical outcomes** | | | |
| ICU duration in days, median, (IQR) | 13 (6–22) | 17 (7–33) | <0.001 |
| Oxygen therapy, n (%) | 152 (96) | 149 (99) | 0.28 |
| Invasive mechanical ventilation, n (%) | 119 (75) | 125 (82) | 0.17 |
| IMV time in days, median, (IQR) | 9 (5–14) | 12 (7–23) | 0.02 |
| Acute kidney injury, n (%) | 119 (75) | 139 (91) | <0.001 |
| In-hospital mortality, n (%) | 44 (28) | 73 (49) | <0.001 |

*Abbreviations*: COPD, chronic obstructive pulmonary disease; AC, anticoagulant; aPTT, activated partial thromboplastin time; ICU, intensive care unit; IMV, invasive mechanical ventilation; PaCO$_2$, partial pressure of carbon dioxide; PaO$_2$, partial pressure of oxygen; pH, potential of hydrogen; IQR, interquartile range.

## Clinical characteristics of the study sample

There were no group differences in demographics, except the prophylactic cohort was younger (median age = 56 years) than the therapeutic cohort (median age = 63 years) (p = 0.002). Primary COVID-19 symptoms at admission were shortness of breath (70–76%), cough (65–68%) and fever (62–69%). Therapeutic AC patients had significantly higher prevalence of COPD (p = 0.005). There were no group differences in vitals at presentation, except for oral temperature, which was lower for therapeutic AC patients (p = 0.004). Between group differences in laboratory values were noted for BNP, LDH, lymphocytes, and $PaO_2$ (p's<0.01), with most extreme values observed for therapeutic dose users. Compared to prophylactic patients, therapeutic patients had significantly higher D-dimer (p<0.001) and higher incidence of DVT (p = 0.002).

## ICU duration, oxygen therapy, AKI and in-hospital mortality

Most ICU patients (>95%) received some form of oxygen therapy. Though there were no group differences in number of patients who received IMV (p = 0.17) or in IMV duration (p>0.01). ICU duration was longer for therapeutic users (median = 17 days) when compared to prophylactic users (median = 13 days, p<0.001). Hospital acquired AKI was significantly higher for therapeutic (91%) than prophylactic users (75%, p<0.001). Uncorrected mortality rate was 28% for prophylactic AC patients, and 49% for therapeutic AC patients (p<0.001).

## Dose-dependent survival analysis: Prophylactic dose vs therapeutic dose

Survival functions of patients who received prophylactic or therapeutic dose anticoagulants are presented in Fig 2. The overall survival probability was not significantly different between groups (log rank test, $\chi^2$ = 3.13, p = 0.077). The overall incidence rate was 15.6 deaths (95% CI = 11.6 to 21.1 deaths) per 1,000 person-days for prophylactic users, and 19.4 deaths (95% CI = 16.1 to 23.3 deaths) per 1,000 person-days for therapeutic users (Table 2). Compared to prophylactic anticoagulated patients, therapeutic anticoagulated patients had non-significantly higher risk of death (adj. HR = 1.21, 95% CI = 0.83 to 1.79, p = 0.32) in Cox regression adjusted for age.

## Risk of death by AC dose and ICU duration

Patients receiving therapeutic dose AC had higher death incidence rates after 3 weeks in the ICU relative to the first 3 weeks, consistent with the survival curve diverging and worsening from prophylactic users at 3 weeks in the ICU (Fig 2). With 3 or more weeks in the ICU, the incidence rate was 2.4 deaths (95% CI = 0.9 to 6.3 deaths) per 1,000 person-days for prophylactic users versus 8.6 deaths (95% CI = 6.2 to 11.9 deaths) per 1,000 person-days for therapeutic users (Table 3). Therapeutic AC patients had nearly 5 times higher risk of death than prophylactic AC patients (adj. HR = 4.89, 95% CI = 1.71 to 14.0, p = 0.003) in Cox regression adjusted for age. This result remained unchanged after including additional covariates to control for clinical status, including thrombotic events (adj. HR = 4.18, 95% CI = 1.44 to 12.1, p = 0.009), and D-dimer and cumulative comorbidities (HR = 5.22, 95% CI = 1.72 to 15.82, p = 0.003).

To better understand what factors were associated with poor survival among therapeutic anticoagulant users with 3 or more weeks of ICU admission, we conducted follow-up analysis comparing survivors (n = 35) and non-survivors (n = 32) in this group of interest (Table 4). Non-survivors were characterized by significantly older age (p = 0.008), higher prevalence of coronary artery disease (p = 0.009), lower lymphocyte count (p = 0.007), and higher incidence of myocardial infarction (p = 0.005). The majority (87%) of non-survivors were male (p = 0.08).

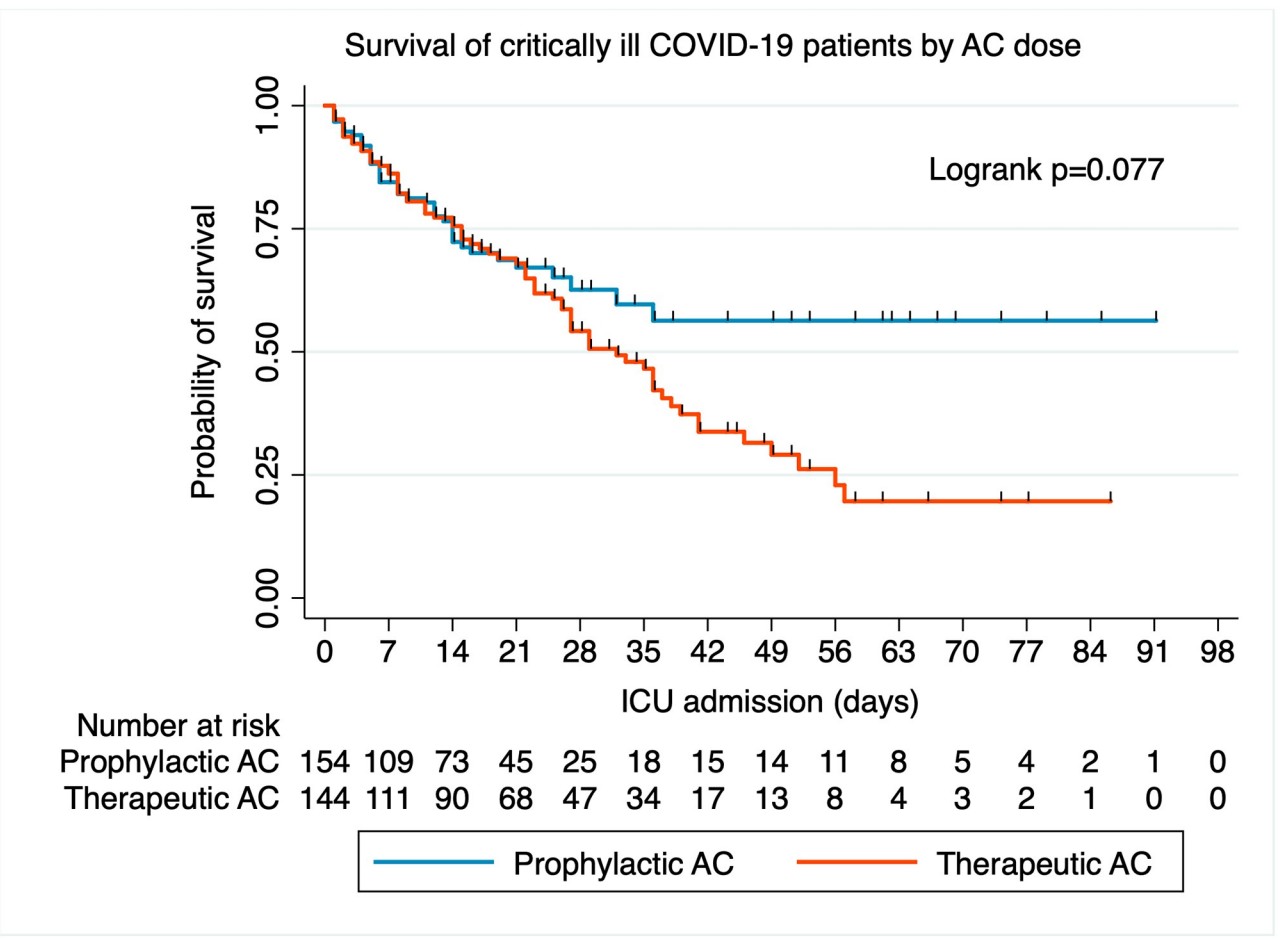

**Fig 2. Kaplan-Meier survival plots by anticoagulant treatment.** The overall survival functions of critically ill COVID-19 patients who received prophylactic (blue line) or therapeutic (red line) dose anticoagulation are not significantly different from each other ($\chi^2 = 3.13$, log rank p = 0.077). However, a clear divergence in survival is noted after 3 weeks ICU admission indicating lower probability of survival among therapeutically anticoagulated patients. Hash marks represent censored data.

**Table 2. Risk of death for ICU patients with COVID-19 by anticoagulant treatment dosage.**

| | Deaths / Person-days | Incidence Rate | | Age-adjusted Hazard Ratio | | |
|---|---|---|---|---|---|---|
| | | Per 1,000 person-days | 95% CI | HR | 95% CI | p-value |
| Prophylactic dose (ref) | 43/2750 | 15.6 | 11.6 to 21.1 | 1.00 | - | - |
| Therapeutic dose | 72/5936 | 19.4 | 16.1 to 23.3 | 1.21 | 0.83 to 1.79 | 0.32 |

*Abbreviations*: CI, confidence interval; HR, hazard ratio.

**Table 3. Risk of death for COVID-19 patients with 3 or more weeks ICU admission by anticoagulant treatment dosage.**

| | Deaths / Person-days | Incidence Rate | | Age-adjusted Hazard Ratio | | |
|---|---|---|---|---|---|---|
| | | Per 1,000 person-days | 95% CI | HR | 95% CI | p-value |
| Prophylactic dose (ref) | 4/1699 | 2.4 | 0.9 to 6.3 | 1.00 | - | - |
| Therapeutic dose | 36/4210 | 8.6 | 6.2 to 11.9 | 4.89 | 1.71 to 14.0 | 0.003 |

*Abbreviations*: CI, confidence interval; HR, hazard ratio.

**Table 4. Sample characteristics of survivors and non-survivors with 21 or more days ICU admission who received therapeutic dose anticoagulation.**

| | Survivors | Non-survivors | p-value |
|---|---|---|---|
| | n = 35 | n = 32 | |
| **Demographics** | | | |
| Age in years, median (IQR) | 57 (50–63) | 67 (59–75) | 0.008 |
| Female sex, n (%) | 11 (31) | 4 (13) | 0.08 |
| **Comorbidities**, n (%) | | | |
| Asthma | 3 (9) | 4 (13) | 0.70 |
| Cancer | 2 (6) | 0 (0) | 0.49 |
| Chronic kidney disease | 3 (9) | 1 (3) | 0.62 |
| COPD | 1 (3) | 3 (9) | 0.34 |
| Coronary artery disease | 0 (0) | 6 (19) | 0.009 |
| Heart failure | 0 (0) | 1 (3) | 0.48 |
| Hypertension | 21 (60) | 18 (56) | 0.81 |
| Immunosuppression | 6 (17) | 2 (6) | 0.26 |
| Type 2 diabetes | 10 (29) | 5 (16) | 0.25 |
| Smoking History | | | 0.50 |
| Current Smoker | 4 (11) | 1 (3) | |
| Former Smoker | 5 (14) | 8 (25) | |
| Never Smoker | 24 (69) | 21 (66) | |
| Unknown | 2 (6) | 2 (6) | |
| **Vital signs at admission**, median (IQR) | | | |
| Heart rate, bpm | 93 (77–102) | 97 (82–104) | 0.45 |
| Diastolic blood pressure, mmHg | 72 (64–77) | 69 (62–75) | 0.46 |
| Respiratory rate, rate/min | 24 (20–32) | 24 (21–30) | 0.96 |
| Oxygen saturation, % | 94 (90–95) | 93 (90–94) | 0.55 |
| Systolic blood pressure, mmHg | 123 (113–133) | 129 (113–142) | 0.45 |
| Temperature, ˚C | 37.2 (37.0–37.9) | 37.1 (36.8–37.5) | 0.07 |
| **Lab values at admission**, median (IQR) | | | |
| Alanine aminotransferase, U/L | 42 (29–69) | 50 (32–73) | 0.43 |
| Aspartate aminotransferase, U/L | 48 (44–75) | 63 (44–89) | 0.25 |
| Bicarbonate, mEg/L | 22 (20–24) | 22 (19–24) | 0.38 |
| C-reactive protein, mg/dL | 12.9 (7.9–25.8) | 17.6 (8.6–28.3) | 0.91 |
| Creatinine mg/dL | 1.1 (0.8–1.6) | 1.0 (0.7–1.3) | 0.29 |
| Ferritin, ng/mL | 1328 (881–1945) | 1458 (764–2054) | 0.93 |
| Hematocrit, % | 39.4 (32.4–45.5) | 40.5 (35.6–43.4) | 0.93 |
| Lactate dehydrogenase, U/L | 565 (420–690) | 623 (438–791) | 0.43 |
| Lymphocytes (lymp/mcL) | 9.3 (6.8–14.8) | 4.4 (2.6–7.9) | 0.007 |
| $paCO_2$, mmHg | 41 (34–53) | 37 (33–41) | 0.25 |
| $paO_2$, mmHg | 80 (67–100) | 62 (55–84) | 0.03 |
| pH | 7.4 (7.4–7.5) | 7.4 (7.4–7.5) | 0.32 |
| Procalcitonin, ng/mL | 0.3 (0.2–0.8) | 0.4 (0.2–0.8) | 0.99 |
| White blood cell count, x$10^3$ /ml | 7.6 (6.3–9.8) | 10.8 (6.2–16.2) | 0.16 |
| **AC parameters at admission**, median, (IQR) | | | |
| D-dimer, ng/mL | 457 (326–892) | 780 (488–2235) | 0.02 |
| Platelet count, $10^9$/L | 196 (169–249) | 207 (149–261) | 0.88 |

(*Continued*)

**Table 4.** (Continued)

| | Survivors | Non-survivors | p-value |
|---|---|---|---|
| | n = 35 | n = 32 | |
| **Escalated care**, n (%) | | | |
| Oxygen therapy | 35 (100) | 32 (100) | >0.99 |
| Invasive mechanical ventilation | 32 (94) | 31 (97) | >0.99 |
| Acute kidney injury | 34 (97) | 32 (100) | >0.99 |

*Abbreviations*: COPD, chronic obstructive pulmonary disease; AC, anticoagulant; aPTT, activated partial thromboplastin time; PaCO$_2$, partial pressure of carbon dioxide; PaO$_2$, partial pressure of oxygen; pH, potential of hydrogen; IQR, interquartile range.

## Discussion

This study described the clinical characteristics and survival probability by anticoagulant treatment of 311 critically ill patients with COVID-19 hospitalized at Stony Brook University Hospital between February 7, 2020, and June 30, 2020. We made the following primary observations: (1) Therapeutic dose AC regimens did not provide a survival benefit over prophylactic dose AC regimens in critically ill COVID-19 patients; (2) The survival curve of the therapeutic cohort diverged from the prophylactic cohort after 3 weeks indicating higher mortality after extended ICU care, which could not be explained by clinical background or thrombotic events alone; (3) Therapeutic dose patients had higher incidence of hospital-acquired AKI; and (4) Non-survivors in the therapeutic cohort with extended ICU care were characterized by older males with depressed lymphocyte count and cardiovascular disease, which represent risk factors that may help identify at-risk patients who require long-term hospitalization with therapeutic dose AC treatment.

Patients therapeutically anticoagulated had a higher prevalence of COPD and presented with high D-dimer, LDH, and BNP values; and low lymphocyte count and PaO$_2$ indicative of more severe COVID-19 disease [25–27], which may explain the high mortality rate among therapeutically anticoagulated patients. In particular, elevated D-dimer level has been associated with COVID-19 disease severity and increased risk of death [28–30], in line with our observations of elevated D-dimer among non-survivors in all groups. As a predictor of thrombotic manifestations of COVID-19, D-dimer might help in early recognition of at-risk patients and also predict outcome. The mechanism responsible for thrombotic events in COVID-19 patients is unclear, but it has been suggested that the inflammatory response to COVID-19 may trigger thrombotic activation in the venous and the arterial circulation [30–32]. This idea is supported by reports of an association between systemic inflammation and increased thrombotic events and bleeding risk in patients without COVID-19 [33–35]. In our study, C-reactive protein, a marker of inflammation, was not significantly increased among non-survivors in the therapeutic AC group. Larger studies are needed to further explore this association.

Few studies explored AC dosage on survival and findings are controversial in populations with severe COVID-19. Overall, survival curves for prophylactic and therapeutic anticoagulated patients were similar, especially during the first 3 weeks of ICU admission. This finding is in line with Nadeem et al. [12] who reported no significant difference in AC dosage on 28-day survival in 149 ICU patients. Lynn and colleagues also showed similar survival curves regardless of dose in ICU patients [15]. In contrast, one study from Sweden by Jonmarker et al. [13] reported high dose AC was associated with lower mortality rates compared to medium and low dose AC in 152 ICU patients. Differences in sample characteristics and COVID-19 disease severity may explain differences in study findings. In particular, the

therapeutic anticoagulated patients in our study had more comorbidities, altered lab values and received more invasive ventilation (83%) than in the study by Jonmarker et al. (54%).

A striking observation is that, after 3 weeks of ICU stay, there was a sharp decline in survival in the therapeutic cohort, showing a marked divergence from the prophylactic cohort. Prolonged treatment with therapeutic dose AC (> 3 weeks) was associated with a nearly 5-fold higher risk of death compared to prophylactic anticoagulated patients, regardless of group differences in age or thrombotic events. To our knowledge, this is the first study to report AC dosage effects by ICU duration in severe COVID-19. In follow-up analysis of therapeutic anticoagulated patients, we found that non-survivors were characterized by older males with heart disease, depressed lymphocytes count at admission, and need for mechanical ventilation, consistent with hypoxia manifestations and poor COVID-19 prognosis [2, 4, 27, 36, 37]. These findings provide novel insights as there is limited survival data on AC regimens in critically ill patients, and these factors may help identify at-risk patients who require long-term hospitalization with therapeutic dose AC treatment.

There are different possible explanations for the higher mortality observed among therapeutically anticoagulated patients in our cohort. Compared to the prophylactic cohort, the therapeutic cohort was older, proportionally more male, and had higher prevalence of preexisting conditions, which are known risk factors for worse COVID-19 outcome [38]. Therapeutic users also had higher incidence of hospital acquired AKI, which is known to be associated with increased risk of COVID-19 related mortality [39]. Whether prolonged use of high dose AC is causally related with higher risk of in-hospital death, possibly mediated by acute kidney injury, requires further studies and randomized controlled trials. While anticoagulants are effective in lowering rates of venous thromboembolism (VTE) [7], they may pose risk for bleeding, which could complicate hospitalization for COVID-19, although randomized trials are needed to confirm this [30]. Over-anticoagulation, linked to profuse glomerular hemorrhage and anticoagulant-related nephropathy (ARN)—a newly recognized form of acute kidney injury [40]—is another concern for hospitalized COVID-19 patients. However, no studies have reported this complication in COVID-19 and, despite high incidence of AKI in the therapeutic cohort, our data is inconclusive regarding ARN as a possible complication of COVID-19 among anticoagulated patients due to lack of data on over-anticoagulation. More studies are needed to explore the possibility of over-anticoagulation as a complication of COVID-19.

This is a retrospective study performed in a single hospital, and therefore would need to be replicated in collaboration with multiple institutions to achieve better generalizability. The Stony Brook data registry recorded whether COVID-19 patients received prophylactic or therapeutic anticoagulation, but additional data on exact dose for each patient was not available. Since we treated the primary exposure as a binary variable (i.e., prophylactic vs therapeutic anticoagulation), the lack of dosage data did not affect the analysis in this study. The results of the present study identify a time-dependent association between survival and anticoagulant treatment regimen among severe COVID-19 patients in the ICU, which might provide useful guidance for future prospective studies that can consider further drug subtype analysis, such as new oral anticoagulants (NOACs) vs coumadin vs heparin effects on COVID-19 outcomes. As with any retrospective study, there could be unintentional patient selection bias, and therefore, randomized controlled trials are needed to assess causal relationships between various AC treatment regimens and COVID-19-related survival and long-term outcomes. Also inherent to cohort studies is the potential for unintentional confounding variables, such as group differences in age, although we attempted to control for this in covariate adjusted analysis.

In conclusion, while it is possible that therapeutic dose AC could simply be associated with more severe COVID-19 disease, our data showed that prolonged ICU admission with

therapeutic dose AC was independently associated with markedly higher mortality rate. Further studies are needed to confirm these findings.

## Author Contributions

**Conceptualization:** Wouter S. Hoogenboom, Joyce Q. Lu, Benjamin Musheyev, Lara Borg, Rebeca Janowicz, Stacey Pamlayne, Wei Hou, Tim Q. Duong.

**Data curation:** Joyce Q. Lu, Benjamin Musheyev, Lara Borg, Rebeca Janowicz, Stacey Pamlayne, Wei Hou.

**Formal analysis:** Wouter S. Hoogenboom, Joyce Q. Lu.

**Methodology:** Wouter S. Hoogenboom, Joyce Q. Lu.

**Supervision:** Wouter S. Hoogenboom, Tim Q. Duong.

**Visualization:** Wouter S. Hoogenboom.

**Writing – original draft:** Wouter S. Hoogenboom, Joyce Q. Lu.

**Writing – review & editing:** Wouter S. Hoogenboom, Tim Q. Duong.

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
