## [Decision Letter · Decision Letter 0]

16 Jun 2021

PONE-D-21-15647

Clinical Characteristics and Survival Analysis of Critically Ill COVID-19 Patients by Anticoagulant Dose

PLOS ONE

Dear Dr. Duong,

Thank you for submitting your manuscript to PLOS ONE. After careful consideration, we feel that it has merit but does not fully meet PLOS ONE’s publication criteria as it currently stands. Therefore, we invite you to submit a revised version of the manuscript that addresses the points raised during the review process.

The manuscript focuses on a topic of current potential interest. However, the study presents several major shortcomings that should be addressed. To mention some of them, i) concern about  the conclusions since this type of study cannot lend support or discourage the use of prophylactic versus therapeutic use of anticoagulants; ii) concern about the fact that the worse clinical background is causing both the poor survival and the selection of the therapeutic regimen; iii) unclear how the equal proportion of therapeutic versus prophylactic regimens was occurring; iv) need to provide data about the dosage of the heparin drip, Eliquis, Xarelto, dabigatran; v) need to highlight more useful messages; vi) unclear the reasons for the higher mortality with therapeutic anticoagulation; vii) concern about the fact that bleeding is not a significant cause of morbidity in hospitalized COVID-19 patients; viii) concern about the fact that anticoagulant-related nephropathy (ARN) is a known concept, but no studies have reported this complication in COVID-19; ix) need to provide data about over-anticoagulation; x) unclear whether there was no selection bias in the groups that received prophylactic versus therapeutic dose anticoagulation; xi) unclear whether older people and those with higher D-dimers were given therapeutic dose.

We look forward to receiving your revised manuscript.

Kind regards,

Giuseppe Remuzzi

Academic Editor

PLOS ONE

Journal Requirements:

Additional Editor Comments (if provided):

Reviewers' comments:

Reviewer's Responses to Questions

**Comments to the Author**

1. Is the manuscript technically sound, and do the data support the conclusions?

Reviewer #1: Yes

Reviewer #2: No

2. Has the statistical analysis been performed appropriately and rigorously? 

Reviewer #1: I Don't Know

Reviewer #2: Yes

3. Have the authors made all data underlying the findings in their manuscript fully available?

Reviewer #1: Yes

Reviewer #2: No

4. Is the manuscript presented in an intelligible fashion and written in standard English?

Reviewer #1: Yes

Reviewer #2: Yes

5. Review Comments to the Author

Reviewer #1: In this paper, the authors try to correlate clinical characteristics and survival in critically ill COVID-19 patients by anticoagulant dosing. The study is interesting and is an addition to several papers in this area. The uniqueness is the longer follow up and focus on critically ill. I have the following comments

Major

1. Although the authors want to say that therapeutic anticoagulation is not ideal, I feel there are more useful messages in here which may be highlighted. For example, the survival curves diverge after three weeks with the two doses. Secondly, there were more kidney injury (not sure how it can be conclusively said to be due to ARN – see below)

2. Can the authors explain what may be the reasons for the higher mortality with therapeutic AC – although you state “investigated the cause of death and found no differences in cause of death between prophylactic and therapeutic groups”

Minor

1. Avoid the term ‘coagulopathy’ which is vague

2. Bleeding is not a significant cause of morbidity in hospitalized COVID-19 patients. Reference 24 does not support this

3. Over-anticoagulation, linked to profuse glomerular hemorrhage and anticoagulant-related nephropathy (ARN)—a newly recognized form of acute kidney injury [25]—is another concern for hospitalized COVID-19 patients. – ARN is a known concept but no studies have reported this complication in COVID-19. If they have been, please quote

4. Also, this paper has not shown any data about over-anticoagulation

5. It is interesting to note that almost equal numbers 158 and 153 received prophylactic and therapeutic dose AC. Please state there was no selection bias. As stated in the demographics, were the older people and those with higher D-dimers were given therapeutic dose?

Reviewer #2: The manuscript entitled "Clinical Characteristics and Survival Analysis of Critically Ill COVID-19 Patients by Anticoagulant Dose" by Duong et al. describes the mortality after anticoagulant therapy in COVID-19 patients.

The topic is of interest, and the results intriguing.

My main critique of the manuscript pertains to the conclusions: this type of study cannot lend support or discourage the use of prophylactic vs. therapeutic use of anticoagulants. The authors can only report the association between the therapeutic AC regimen and the poor survival. The authors can only speculate why this association is present in the discussion section. Indeed, as the authors hypothesize, one possibility is that the therapeutic AC regimen is bad.

However, it is equally possible that patients treated at high dose AC (heparin drip, Eliquis / apixaban, Xarelto / rivaroxaban, dabigatran / Pradaxa, warfarin / coumadin, enoxaparin / Lovenox 1mg / kg twice or 1.5mg / kg) were also those with an initial bad clinical background. Indeed, these patients had greater D-dimer values and COPD. In this scenario, the worse clinical background is causing both the poor survival and the selection of the therapeutic regimen.

I also noticed that the number of patients treated with enoxaparin 40mg twice was very much comparable with that of enoxaparin 1 mg/kg twice. This is intriguing as doctors apparently selected the AC regimen at random (if we hypothesize that the two groups of patients have overlapping clinical characteristics). I would ask the authors to explain how this equal proportion of therapeutic vs. prophylactic regimens was occurring.

Minor points:

No data are reported about the dosage of the heparin drip, Eliquis, Xarelto, dabigatran: please describe this info better.

In the list of drugs, sometimes you use first the commercial and then the official drug name, and sometimes the other way round (e.g., Eliquis / apixaban, Xarelto / rivaroxaban, dabigatran / Pradaxa, warfarin/coumadin). Please be consistent.

6. PLOS authors have the option to publish the peer review history of their article (what does this mean?). If published, this will include your full peer review and any attached files.

Reviewer #1: No

Reviewer #2: No

---

## [Author Response · Author response to Decision Letter 0]

13 Aug 2021

Academic Editor

1. Concern about the conclusions since this type of study cannot lend support or discourage the use of prophylactic versus therapeutic use of anticoagulants;

Response: We agree that the observational nature of our study does not allow recommendations regarding anticoagulant treatment. Therefore, we modified the conclusions throughout the manuscript:

Abstract:

“Our findings raise the possibility that prolonged use of high dose AC, independent of thrombotic events or clinical background, might be associated with higher risk of in-hospital mortality., and do not support routine and long-term use of therapeutic dose AC in COVID-19 patients admitted to the ICU.”

Discussion (first paragraph):

“This study described the clinical characteristics and survival probability by anticoagulant treatment of 311 critically ill patients with COVID-19 hospitalized at Stony Brook University Hospital between February 7, 2020, and June 30, 2020. . . Our findings do not support routine long-term use of therapeutic dose AC in unselected patients with severe COVID-19.”

2. Concern about the fact that the worse clinical background is causing both the poor survival and the selection of the therapeutic regimen;

Response: In our cohort, only COPD was significantly more prevalent in the therapeutic group. We performed additional analysis with COPD and cumulative comorbidities (i.e., hypertension, diabetes, asthma, coronary artery disease, COPD, heart failure, cancer, immunosuppression, and chronic kidney disease) as additional covariates to control for clinical background. We found that including these covariates did not change the results. That is, the therapeutically anticoagulated cohort with 3 or more weeks of ICU admission had significantly higher risk of death compared to the prophylactically anticoagulated cohort in Cox proportional hazard model with age and comorbidities as covariates:

With age and COPD as covariates: adj. HR = 5.04, 95% CI = 1.75 to 14.51, p = 0.003

With age and cumulative comorbidities as covariates: adj. HR = 4.82, 95% CI = 1.66 to 13.89, p = 0.004.

It is likely that elevated D-dimer at admission contributed to the selection of therapeutic regimen, but D-dimer and age as covariates in Cox regression did not change the results (HR = 4.71, 95% CI = 1.61 to 13.77, p = 0.005), nor did D-dimer, comorbidities and age as covariates (HR = 5.22, 95% CI = 1.72 to 15.82, p = 0.003). 

We included this additional analysis in the Results section on page 11 to support our findings:

“Therapeutic AC patients had nearly 5 times higher risk of death than prophylactic AC patients (adj. HR = 4.89, 95% CI = 1.71 to 14.0, p = 0.003) in Cox regression adjusted for age. This result remained unchanged after including additional covariates to control for clinical status, including thrombotic events (adj. HR = 4.18, 95% CI = 1.44 to 12.1, p = 0.009), and D-dimer and cumulative comorbidities (HR = 5.22, 95% CI = 1.72 to 15.82, p = 0.003).”

3. Unclear how the equal proportion of therapeutic versus prophylactic regimens was occurring; 

Response: The approximately equal group proportions were not predetermined, but a coincidental result after applying study exclusion criteria and grouping patients by AC dose, as described in more detail in the ‘Study Population and data collection’ of the Methods section (page 6) and illustrated in Figure 1. 

4. Need to provide data about the dosage of the heparin drip, Eliquis, Xarelto, dabigatran; 

Response: Anticoagulant dosage varied in time and across patients. Dosage was continuous in some patients and changing in others. Patients were grouped as low dose (continuous low dosage only) or high dose (continuous high dose or change to high dose) based on the criteria described in the text. We revised text in the Methods section (page 5) to clarify patient grouping based on dosage:

“Patients were divided into one of two groups based on AC treatment dosage received: (1) continuous and exclusively low dose / prophylactic anticoagulation – that is, subcutaneous heparin or Lovenox (enoxaparin) at a dose of 40 mg twice daily; or (2) high dose / therapeutic anticoagulation – that is, 1 mg/kg twice daily or 1.5 mg/kg daily of the following anticoagulants: any heparin drip, Eliquis (apixaban), Xarelto (rivaroxaban), Pradaxa (dabigatran), Coumadin (warfarin), or Lovenox (enoxaparin)). Patients who received both prophylactic and therapeutic dosages during their hospital stay were grouped in the therapeutic cohort for this analysis.”

5. Need to highlight more useful messages; 

Response: We agree and amended the text to clarify and highlight messages. Of note, due to the observational nature of this study, we can only report the existence of associations between exposure and outcomes:

Abstract (page 2, Results):

“For the first 3 weeks of ICU stay, we observed similar survival curves for prophylactic and therapeutic AC groups. However, after 3 or more weeks of ICU stay, the therapeutic AC group, characterized by high incidence of acute kidney injury, had markedly higher death incidence rates with 8.6 deaths (95% CI=6.2-11.9 deaths) per 1,000 person-days and about 5 times higher risk of death (adj. HR=4.89, 95% CI=1.71-14.0, p=0.003) than the prophylactic group (2.4 deaths [95% CI=0.9–6.3 deaths] per 1,000 person-days). Non-survivors were characterized by older males with depressed lymphocyte counts and cardiovascular disease.”

Abstract (page 2, Conclusions):

“Our findings raise the possibility that prolonged use of high dose AC, independent of thrombotic events or clinical background, might be associated with higher risk of in-hospital mortality. Moreover, AKI, age, lymphocyte count, and cardiovascular disease may represent important risk factors that could help identify at-risk patients who require long-term hospitalization with therapeutic dose AC treatment.”

Discussion (page 12):

“We made the following primary study observations: (1) Therapeutic dose AC regimens did not provide a survival benefit over prophylactic dose AC regimens in critically ill COVID-19 patients; (2) The survival curve of the therapeutic cohort diverged from the prophylactic cohort after 3 weeks indicating higher mortality after extended ICU care, which could not be explained by clinical background or thrombotic events alone; (3) Therapeutic dose patients had higher incidence of hospital-acquired AKI; and (4) Non-survivors in the therapeutic cohort with extended ICU care were characterized by older males with depressed lymphocyte count and cardiovascular disease, which represent risk factors that may help identify at-risk patients who require long-term hospitalization with therapeutic dose AC treatment.”

6. Unclear the reasons for the higher mortality with therapeutic anticoagulation

Response: We expanded the discussion on possible reasons for the higher mortality with therapeutic anticoagulation (page 14):

“There are different possible explanations for the higher mortality observed among therapeutically anticoagulated patients in our cohort. Compared to the prophylactic cohort, the therapeutic cohort was older, proportionally more male, and had higher prevalence of preexisting conditions, which are known risk factors for worse COVID-19 outcomes (Parohan et al., 2020). Therapeutic users also had higher incidence of hospital acquired AKI, which is known to be associated with increased risk of COVID-19 related mortality (Chan et al., 2021). Whether prolonged use of high dose AC is causally related with higher risk of in-hospital death, possibly mediated by acute kidney injury, requires further studies and randomized controlled trials.”

New references added:

Parohan M, Yaghoubi S, Seraji A, Javanbakht MH, Sarraf P, Djalali M. Risk factors for mortality in patients with Coronavirus disease 2019 (COVID-19) infection: a systematic review and meta-analysis of observational studies. Aging Male. 2020 Dec;23(5):1416-1424. doi: 10.1080/13685538.2020.1774748. Epub 2020 Jun 8. PMID: 32508193.

Chan L, Chaudhary K, Saha A, Chauhan K, Vaid A, Zhao S, Paranjpe I, Somani S, Richter F, Miotto R, Lala A, Kia A, Timsina P, Li L, Freeman R, Chen R, Narula J, Just AC, Horowitz C, Fayad Z, Cordon-Cardo C, Schadt E, Levin MA, Reich DL, Fuster V, Murphy B, He JC, Charney AW, Böttinger EP, Glicksberg BS, Coca SG, Nadkarni GN; Mount Sinai COVID Informatics Center (MSCIC). AKI in Hospitalized Patients with COVID-19. J Am Soc Nephrol. 2021 Jan;32(1):151-160. doi: 10.1681/ASN.2020050615. Epub 2020 Sep 3. PMID: 32883700; PMCID: PMC7894657.

7. Concern about the fact that bleeding is not a significant cause of morbidity in hospitalized COVID-19 patients; 

Response: Patients with bleeding events did not receive anticoagulation or required a pause in anticoagulation during hospital stay and were excluded from analysis. Of the remaining patients, we are unable to provide data about other significant bleeding events because if present they were not significant enough to warrant discontinuation of anticoagulation. We updated the methods and discussion section to clarify the text:

Methods section (page 5)

“Patients with contraindications for anticoagulant treatment who did not receive anticoagulation or who required a pause in anticoagulation during hospital stay were excluded from the analysis.”

Discussion (page 14):

“While anticoagulants are effective in lowering rates of venous thromboembolism (VTE) [7], they may pose risk for bleeding, which could complicate hospitalization for COVID-19, although randomized trials are needed to confirm this [24].”

8. Concern about the fact that anticoagulant-related nephropathy (ARN) is a known concept, but no studies have reported this complication in COVID-19; 

Response: We agree and modified the discussion (page 14) to read:

“Over-anticoagulation, linked to profuse glomerular hemorrhage and anticoagulant-related nephropathy (ARN)—a newly recognized form of acute kidney injury [25]—is another concern for hospitalized COVID-19 patients. However, no studies have reported this complication in COVID-19 and, despite high incidence of AKI in the therapeutic cohort, our data is inconclusive regarding ARN as a possible complication of COVID-19 among anticoagulated patients due to lack of data on over-anticoagulation. More studies are needed to explore the possibility of over-anticoagulation as a complication of COVID-19.”

9. Need to provide data about over-anticoagulation;

Response: The status of overanticoagulation could not be determined from available data in the patient charts. Please see previous point 8 for a modified discussion. 

10. Unclear whether there was no selection bias in the groups that received prophylactic versus therapeutic dose anticoagulation; 

Response: Dose regimens were based on general risk assessment considering clinical background, preexisting conditions and presenting laboratory values. Furthermore, potential confounder (e.g., age, d-dimer and comorbidities) were used as covariates in analysis and now described in the Results (page 10). We further acknowledge that unintentional patient selection bias is inherent to retrospective studies and suggested that randomized controlled trials are needed to assess causal relationships. We expanded the Methods and Limitations section to read:

Methods section (page 5):

“Dose regimens were based on general risk assessment considering clinical background, preexisting conditions, and presenting laboratory values.”

Limitations section (page 14):

“As with any retrospective study, there could be unintentional patient selection bias, and therefore, randomized controlled trials are needed to assess causal relationships between various AC treatment regimens and COVID-19-related survival and long-term outcomes. Also inherent to observational studies is the potential for unintentional confounding variables, such as group differences in age, although we attempted to control for this in covariate adjusted analysis.”

11. Unclear whether older people and those with higher D-dimers were given therapeutic dose.

Response: As indicated in point 10, dose regimens were based on general risk assessment considering clinical background, preexisting conditions, and presenting laboratory values. Furthermore, potential confounders such as age and d-dimer were used as covariates in analysis. We amended the Methods (page 5) to read:

“Dose regimens were based on general risk assessment considering clinical background, preexisting conditions, and presenting laboratory values.”

Response: We revisited the PLOS ONE style and submission guidelines and updated the manuscript accordingly.

2. We note that you have indicated that data from this study are available upon request. PLOS only allows data to be available upon request if there are legal or ethical restrictions on sharing data publicly. 

Response: There are no ethical or legal restrictions on sharing a de-identified data set. We will make available an anonymized data set necessary to replicate our study findings. 

 

Reviewer 1

Major

1. Although the authors want to say that therapeutic anticoagulation is not ideal, I feel there are more useful messages in here which may be highlighted. For example, the survival curves diverge after three weeks with the two doses. Secondly, there were more kidney injury (not sure how it can be conclusively said to be due to ARN – see below)

Response: We agree and amended the text to clarify and highlight messages. Of note, due to the observational nature of this study, we can only report the existence of associations between exposure and outcomes:

Abstract (page 2, Results):

“For the first 3 weeks of ICU stay, we observed similar survival curves for prophylactic and therapeutic AC groups. However, after 3 or more weeks of ICU stay, the therapeutic AC group, characterized by high incidence of acute kidney injury, had markedly higher death incidence rates with 8.6 deaths (95% CI=6.2-11.9 deaths) per 1,000 person-days and about 5 times higher risk of death (adj. HR=4.89, 95% CI=1.71-14.0, p=0.003) than the prophylactic group (2.4 deaths [95% CI=0.9–6.3 deaths] per 1,000 person-days). Non-survivors were characterized by older males with depressed lymphocyte counts and cardiovascular disease.”

Abstract (page 2, Conclusions):

“Our findings raise the possibility that prolonged use of high dose AC, independent of thrombotic events or clinical background, might be associated with higher risk of in-hospital mortality. Moreover, AKI, age, lymphocyte count, and cardiovascular disease may represent important risk factors that could help identify at-risk patients who require long-term hospitalization with therapeutic dose AC treatment.”

Discussion (page 12):

“We made the following primary observations: (1) Therapeutic dose AC regimens did not provide a survival benefit over prophylactic dose AC regimens in critically ill COVID-19 patients; (2) The survival curve of the therapeutic cohort diverged from the prophylactic cohort after 3 weeks indicating higher mortality after extended ICU care, which could not be explained by clinical background or thrombotic events alone; (3) Therapeutic dose patients had higher incidence of hospital-acquired AKI; and (4) Non-survivors in the therapeutic cohort with extended ICU care were characterized by older males with depressed lymphocyte count and cardiovascular disease, which represent risk factors that may help identify at-risk patients who require long-term hospitalization with therapeutic dose AC treatment.”

2. Can the authors explain what may be the reasons for the higher mortality with therapeutic AC – although you state “investigated the cause of death and found no differences in cause of death between prophylactic and therapeutic groups”

Response: We expanded the discussion on possible reasons for the higher mortality with therapeutic anticoagulation (page 14):

“There are different possible explanations for the higher mortality observed among therapeutically anticoagulated patients in our cohort. Compared to the prophylactic cohort, the therapeutic cohort was older, proportionally more male, and had higher prevalence of preexisting conditions, which are known risk factors for worse COVID-19 outcomes (Parohan et al., 2020). Therapeutic users also had higher incidence of hospital acquired AKI, which is known to be associated with increased risk of COVID-19 related mortality (Chan et al., 2021). Whether prolonged use of high dose AC is causally related with higher risk of in-hospital death, possibly mediated by acute kidney injury, requires further studies and randomized controlled trials.”

New references added:

Parohan M, Yaghoubi S, Seraji A, Javanbakht MH, Sarraf P, Djalali M. Risk factors for mortality in patients with Coronavirus disease 2019 (COVID-19) infection: a systematic review and meta-analysis of observational studies. Aging Male. 2020 Dec;23(5):1416-1424. doi: 10.1080/13685538.2020.1774748. Epub 2020 Jun 8. PMID: 32508193.

Chan L, Chaudhary K, Saha A, Chauhan K, Vaid A, Zhao S, Paranjpe I, Somani S, Richter F, Miotto R, Lala A, Kia A, Timsina P, Li L, Freeman R, Chen R, Narula J, Just AC, Horowitz C, Fayad Z, Cordon-Cardo C, Schadt E, Levin MA, Reich DL, Fuster V, Murphy B, He JC, Charney AW, Böttinger EP, Glicksberg BS, Coca SG, Nadkarni GN; Mount Sinai COVID Informatics Center (MSCIC). AKI in Hospitalized Patients with COVID-19. J Am Soc Nephrol. 2021 Jan;32(1):151-160. doi: 10.1681/ASN.2020050615. Epub 2020 Sep 3. PMID: 32883700; PMCID: PMC7894657.

Minor

1. Avoid the term ‘coagulopathy’ which is vague

Response: Thank you for this suggestion. We removed the term from the manuscript. 

2. Bleeding is not a significant cause of morbidity in hospitalized COVID-19 patients. Reference 24 does not support this

Response: We revised the text to read: 

“While anticoagulants are effective in lowering rates of venous thromboembolism (VTE) [7], they may pose risk for bleeding and are a significant cause of morbidity in hospitalized COVID-19 patients, which could complicate hospitalization for COVID-19, although this needs to be confirmed in randomized trials [24].”

3. “Over-anticoagulation, linked to profuse glomerular hemorrhage and anticoagulant-related nephropathy (ARN)—a newly recognized form of acute kidney injury [25]—is another concern for hospitalized COVID-19 patients.” – ARN is a known concept but no studies have reported this complication in COVID-19. If they have been, please quote

Response: We agree and only speculated that over-anticoagulation could be a possible complication of COVID-19 that should be explored. We modified the discussion (page 14) to read:

“Over-anticoagulation, linked to profuse glomerular hemorrhage and anticoagulant-related nephropathy (ARN)—a newly recognized form of acute kidney injury [25]—is another concern for hospitalized COVID-19 patients. However, no studies have reported this complication in COVID-19 and, despite high incidence of AKI in the therapeutic cohort, our data is inconclusive regarding ARN as a possible complication of COVID-19 among anticoagulated patients due to lack of data on over-anticoagulation. More studies are needed to explore the possibility of over-anticoagulation as a complication of COVID-19.”

4. Also, this paper has not shown any data about over-anticoagulation

Response: The status of overanticoagulation could not be determined from available data in the patient charts. Please see previous point 3 for a modified discussion. 

5. It is interesting to note that almost equal numbers 158 and 153 received prophylactic and therapeutic dose AC. Please state there was no selection bias. As stated in the demographics, were the older people and those with higher D-dimers were given therapeutic dose?

Response: The approximately equal group proportions were not predetermined, but a coincidental result after applying study exclusion criteria and grouping patients by AC dose, as described in more detail on page 6 and illustrated in Figure 1. 

 

Reviewer 2

The manuscript entitled "Clinical Characteristics and Survival Analysis of Critically Ill COVID-19 Patients by Anticoagulant Dose" by Duong et al. describes the mortality after anticoagulant therapy in COVID-19 patients. The topic is of interest, and the results intriguing.

1. My main critique of the manuscript pertains to the conclusions: this type of study cannot lend support or discourage the use of prophylactic vs. therapeutic use of anticoagulants. The authors can only report the association between the therapeutic AC regimen and the poor survival. The authors can only speculate why this association is present in the discussion section. Indeed, as the authors hypothesize, one possibility is that the therapeutic AC regimen is bad.

Response: Thank you for pointing out this important aspect. We agree that the observational nature of our study does not allow recommendations regarding anticoagulant treatment. Therefore, we modified the conclusions throughout the manuscript:

Abstract:

“Our findings raise the possibility that prolonged use of high dose AC, independent of thrombotic events or clinical background, might be associated with higher risk of in-hospital mortality., and do not support routine and long-term use of therapeutic dose AC in COVID-19 patients admitted to the ICU.”

Discussion (first paragraph):

“This study described the clinical characteristics and survival probability by anticoagulant treatment of 311 critically ill patients with COVID-19 hospitalized at Stony Brook University Hospital between February 7, 2020, and June 30, 2020. . . Our findings do not support routine long-term use of therapeutic dose AC in unselected patients with severe COVID-19.”

2. However, it is equally possible that patients treated at high dose AC (heparin drip, Eliquis / apixaban, Xarelto / rivaroxaban, dabigatran / Pradaxa, warfarin / coumadin, enoxaparin / Lovenox 1mg / kg twice or 1.5mg / kg) were also those with an initial bad clinical background. Indeed, these patients had greater D-dimer values and COPD. In this scenario, the worse clinical background is causing both the poor survival and the selection of the therapeutic regimen.

Response: In our cohort, only COPD was significantly more prevalent in the therapeutic group. We performed additional analysis with COPD and cumulative comorbidities (i.e., hypertension, diabetes, asthma, coronary artery disease, COPD, heart failure, cancer, immunosuppression, and chronic kidney disease) as additional covariates to control for clinical background. We found that including these covariates did not change the results. That is, the therapeutically anticoagulated cohort with 3 or more weeks of ICU admission had significantly higher risk of death compared to the prophylactically anticoagulated cohort in Cox proportional hazard model with age and comorbidities as covariates:

With age and COPD as covariates: adj. HR = 5.04, 95% CI = 1.75 to 14.51, p = 0.003

With age and cumulative comorbidities as covariates: adj. HR = 4.82, 95% CI = 1.66 to 13.89, p = 0.004.

It is likely that elevated D-dimer at admission contributed to the selection of therapeutic regimen, but D-dimer and age as covariates in Cox regression did not change the results (HR = 4.71, 95% CI = 1.61 to 13.77, p = 0.005), nor did D-dimer, comorbidities and age as covariates (HR = 5.22, 95% CI = 1.72 to 15.82, p = 0.003). 

We included this additional analysis in the Results section on page 11 to support our findings:

“Therapeutic AC patients had nearly 5 times higher risk of death than prophylactic AC patients (adj. HR = 4.89, 95% CI = 1.71 to 14.0, p = 0.003) in Cox regression adjusted for age. This result remained unchanged after including additional covariates to control for clinical status, including thrombotic events (adj. HR = 4.18, 95% CI = 1.44 to 12.1, p = 0.009), and D-dimer and cumulative comorbidities (HR = 5.22, 95% CI = 1.72 to 15.82, p = 0.003).”

3. I also noticed that the number of patients treated with enoxaparin 40mg twice was very much comparable with that of enoxaparin 1 mg/kg twice. This is intriguing as doctors apparently selected the AC regimen at random (if we hypothesize that the two groups of patients have overlapping clinical characteristics). I would ask the authors to explain how this equal proportion of therapeutic vs. prophylactic regimens was occurring.

Response: The approximately equal group proportions were not predetermined, but a coincidental result after applying study exclusion criteria and grouping patients by AC dose, as described in more detail in the ‘Study Population and data collection’ of the Methods section (page 6) and illustrated in Figure 1.

We added clarification on how dose regimens were determined by the treating clinicians. Methods section (page 5):

“Dose regimens were based on general risk assessment considering clinical background, preexisting conditions, and presenting laboratory values.”

We further acknowledge the existence of unintentional patient selection bias inherent to retrospective studies. Limitations section (page 14):

“As with any retrospective study, there could be unintentional patient selection bias, and therefore, randomized controlled trials are needed to assess causal relationships between various AC treatment regimens and COVID-19-related survival and long-term outcomes. Also inherent to observational studies is the potential for unintentional confounding variables, such as group differences in age, although we attempted to control for this in covariate adjusted analysis.”

Minor points:

4. No data are reported about the dosage of the heparin drip, Eliquis, Xarelto, dabigatran: please describe this info better.

Response: Anticoagulant dosage varied in time and across patients. Dosage was continuous in some patients and changing in others. Patients were grouped as low dose (continuous low dosage only) or high dose (continuous high dose or change to high dose) based on the criteria described in the text. We revised text in the Methods section (page 5) to clarify patient grouping based on dosage:

“Patients were divided into one of two groups based on AC treatment dosage received: (1) continuous and exclusively low dose / prophylactic anticoagulation – that is, subcutaneous heparin or Lovenox (enoxaparin) at a dose of 40 mg twice daily; or (2) high dose / therapeutic anticoagulation – that is, 1 mg/kg twice daily or 1.5 mg/kg daily of the following anticoagulants: any heparin drip, Eliquis (apixaban), Xarelto (rivaroxaban), Pradaxa (dabigatran), Coumadin (warfarin), or Lovenox (enoxaparin)). Patients who received both prophylactic and therapeutic dosages during their hospital stay were grouped in the therapeutic cohort for this analysis.”

5. In the list of drugs, sometimes you use first the commercial and then the official drug name, and sometimes the other way round (e.g., Eliquis / apixaban, Xarelto / rivaroxaban, dabigatran / Pradaxa, warfarin/coumadin). Please be consistent.

Response: Thank you for pointing this out. We revised the text to be consistent:

“Patients were divided into one of two groups based on AC treatment dosage received: (1) continuous and exclusively low dose / prophylactic anticoagulation – that is, subcutaneous heparin or Lovenox (enoxaparin) at a dose of 40 mg twice daily; or (2) high dose / therapeutic anticoagulation – that is, 1 mg/kg twice daily or 1.5 mg/kg daily of the following anticoagulants: any heparin drip, Eliquis (apixaban), Xarelto (rivaroxaban), Pradaxa (dabigatran), Coumadin (warfarin), or Lovenox (enoxaparin). Patients who received both prophylactic and therapeutic dosages during their hospital stay were grouped in the therapeutic cohort for this analysis.”

---

## [Decision Letter · Decision Letter 1]

6 Oct 2021

PONE-D-21-15647R1Dose and duration effects of anticoagulant treatment among critically ill COVID-19 patientsPLOS ONE

Dear Dr. Duong,

Thank you for submitting your manuscript to PLOS ONE. After careful consideration, we feel that it has merit but does not fully meet PLOS ONE’s publication criteria as it currently stands. Therefore, we invite you to submit a revised version of the manuscript that addresses the points raised during the review process.

**The revised manuscript is improved. Nevertheless, few additional issues remain to be addressed.**

**To mention some of them, i) unclear the statement that any heparin drip should be delivered at 1 mg/kg (page 5); ii) concern about the fact that coumadin, eliquis, xarelto can be administered at 1 mg/kg twice a day (Fig 5); iii) need to consider that coumadin might give a procoagulant effect initially; iv) need to report the PTT in the heparin group: v) unclear why the authors do not report a table describing survival parameters also for the group without therapeutic anticoagulation.**

We look forward to receiving your revised manuscript.

Kind regards,

Giuseppe Remuzzi

Academic Editor

PLOS ONE

Journal Requirements:

Additional Editor Comments (if provided):

Reviewers' comments:

Reviewer's Responses to Questions

**Comments to the Author**

1. If the authors have adequately addressed your comments raised in a previous round of review and you feel that this manuscript is now acceptable for publication, you may indicate that here to bypass the “Comments to the Author” section, enter your conflict of interest statement in the “Confidential to Editor” section, and submit your "Accept" recommendation.

Reviewer #1: All comments have been addressed

Reviewer #2: All comments have been addressed

2. Is the manuscript technically sound, and do the data support the conclusions?

Reviewer #1: Yes

Reviewer #2: Yes

3. Has the statistical analysis been performed appropriately and rigorously? 

Reviewer #1: I Don't Know

Reviewer #2: Yes

4. Have the authors made all data underlying the findings in their manuscript fully available?

Reviewer #1: Yes

Reviewer #2: No

5. Is the manuscript presented in an intelligible fashion and written in standard English?

Reviewer #1: Yes

Reviewer #2: Yes

6. Review Comments to the Author

Reviewer #1: All the comments have been adequately addressed. All the important aspects have been taken into consideration. No further changes are needed

Reviewer #2: In the revised manuscript the authors agree that the observational design used was not sufficient to discourage the use of prophylactic AC because of potential selection bias of the treatment and control groups.

I still find it confusing stating that any heparin drip should be delivered at 1mg/kg (Pag 5)

Similarly, are you sure whether coumadin, eliquis,xarelto can be administered at 1mg/kg twice a day (fig 5)? If not, please specify which dose you have used.

Furthermore, before collecting data from NOACs, coumadin, and heparin users as if they are the same therapy, you should consider that

1) coumadin might give a procoagulant effect initially: if this group was worse than the others, maybe this should be taken into consideration (without heparin bridge). The INR should be reported in this group and if it was lower than the therapeutic range these patients might even be considered controls

2) in the heparin group, the PTT should be reported, and you should verify it was in the therapeutic range (see above)

Indeed, in the therapeutic anticoagulation group, survival was linked to lower D-dimer, suggesting that some coagulation effect was present (Table 4).

We have only a Table describing survival parameters in the therapeutic dose group. Why you do not report a similar Table for the group without therapeutic anticoagulation?

7. PLOS authors have the option to publish the peer review history of their article (what does this mean?). If published, this will include your full peer review and any attached files.

Reviewer #1: **Yes: **Jecko Thachil

Reviewer #2: No

---

## [Author Response · Author response to Decision Letter 1]

14 Oct 2021

Comments to the Author

Reviewer #1: All the comments have been adequately addressed. All the important aspects have been taken into consideration. No further changes are needed

Reviewer #2: In the revised manuscript the authors agree that the observational design used was not sufficient to discourage the use of prophylactic AC because of potential selection bias of the treatment and control groups.

1. I still find it confusing stating that any heparin drip should be delivered at 1mg/kg (Pag 5) Similarly, are you sure whether coumadin, eliquis,xarelto can be administered at 1mg/kg twice a day (fig 5)? If not, please specify which dose you have used.

Response: Thank you for pointing this out and we apologize for the confusion. We revisited the Stony Brook database registry for COVID-19 positive patients and associated data codebooks to verify anticoagulant information recorded, and made the following corrections on page 5:

“Patients were divided into one of two groups based on AC treatment dosage received: (1) continuous and exclusively low dose / prophylactic anticoagulation – that is, subcutaneous heparin or Lovenox (enoxaparin) at a dose of 40 mg twice daily; or (2) high dose / therapeutic anticoagulation – that is a) any heparin drip; b) Eliquis (apixaban), Xarelto (rivaroxaban), Pradaxa (dabigatran), or Coumadin (warfarin) at a typical therapeutic dose strength; or c) Lovenox (enoxaparin) at a dose of 1 mg/kg twice daily or 1.5 mg/kg daily.”

Of note, exact dosage regimens for Eliquis (apixaban), Xarelto (rivaroxaban), Pradaxa (dabigatran), and Coumadin (warfarin) were not recorded in the registry, only that they were given for therapeutic anticoagulation. For our study, it was sufficient to know whether someone received prophylactic or therapeutic anticoagulation, which was clearly recorded in the registry. Due to the retrospective nature of this study, additional data on dose of anticoagulation for each patient is not available, which we believe does not affect the results or conclusions of our study, but it is a study limitation that we acknowledge in the discussion (page 14):

“This is a retrospective study performed in a single hospital, and therefore would need to be replicated in collaboration with multiple institutions to achieve better generalizability. The Stony Brook data registry recorded whether COVID-19 patients received prophylactic or therapeutic anticoagulation, but additional data on exact dose for each patient was not available. Since we treated the primary exposure as a binary variable (i.e., prophylactic vs therapeutic anticoagulation), the lack of dosage data did not affect the analysis in this study. The results of the present study identify a time-dependent association between survival and anticoagulant treatment regimen among severe COVID-19 patients in the ICU, which might provide useful guidance for future prospective studies that can consider further drug subtype analysis, such as new oral anticoagulants (NOACs) vs coumadin vs heparin effects on COVID-19 outcomes. As with any retrospective study, there could be unintentional patient selection bias, and therefore, randomized controlled trials are needed to assess causal relationships between various AC treatment regimens and COVID-19-related survival and long-term outcomes. Also inherent to cohort studies is the potential for unintentional confounding variables, such as group differences in age, although we attempted to control for this in covariate adjusted analysis.”

2. Furthermore, before collecting data from NOACs, coumadin, and heparin users as if they are the same therapy, you should consider that 1) coumadin might give a procoagulant effect initially: if this group was worse than the others, maybe this should be taken into consideration (without heparin bridge). The INR should be reported in this group and if it was lower than the therapeutic range these patients might even be considered controls

Response: Thank you for your suggestions. While we agree that coumadin may have a temporary procoagulant effect when started without a heparin bridge, most patients that were on coumadin had pre-existing conditions such as Afib and were already taking prophylactic coumadin. Therefore, we believe that any procoagulant effects among coumadin users in our sample are limited. Of note, only 7 individuals (4.6%) in our study used coumadin at any time during hospitalization, and the small number of this group is unlikely to change the results of this study. Finally, we further note that stratified analysis by anticoagulation drug (i.e., NOACs vs coumadin vs heparin) is an interesting research venue, but beyond the focus of the current study.

3. In the heparin group, the PTT should be reported, and you should verify it was in the therapeutic range (see above)

Response: Lab values in this retrospective study were collected at admission. For this reason, we believe PTT values would not be representative as they were collected before AC treatment for most people. Due to the retrospective nature of this study, unfortunately, we cannot collect additional information, which is a study limitation inherent to the retrospective design of this study, which we acknowledge in the discussion. Of note, if a patient would show a PTT associated with subtherapeutic range, the AC regimen would be adjusted according to hospital protocol. Therefore, despite lack of PTT data, we believe that clinical management would ensure therapeutic range for heparin users. We made the following clarifications in the text:

To indicate the timing when lab values were collected (page 6):

“We collected COVID-19 outcome variables that included mortality rate, ICU duration, oxygen therapy, invasive mechanical ventilator (IMV) use, and incidence of acute kidney injury (AKI). We also collected demographics (i.e., age, gender, ethnicity and race), chronic comorbidities (i.e., smoking history, diabetes, hypertension, asthma, chronic obstructive pulmonary disease (COPD), coronary artery disease (CAD), heart failure, cancer, immune- suppression, chronic kidney disease (CKD)), vitals (i.e., heart rate (HR), diastolic blood pressure (DBP), systolic blood pressure (SBP), respiratory rate, pulse oxygen saturation (SpO2) and temperature), coagulation parameters (D-dimer, platelet count) and other laboratory tests (i.e., creatinine, procalcitonin, aspartate transaminase (AST), alanine aminotransferase (ALT), ferritin, lactate dehydrogenase (LDH), white blood cell count (WBC), C-reactive protein (CRP), lymphocytes count, brain natriuretic peptide (BNP), albumin, hematocrit, bicarbonate, creatinine), and blood gas variables (i.e., pO2, pCO2, and pH). All patient characteristics, symptoms, vitals, and laboratory values were collected at admission.”

And to highlight important future directions (page 14):

“This is a retrospective study performed in a single hospital, and therefore would need to be replicated in collaboration with multiple institutions to achieve better generalizability. The Stony Brook data registry recorded whether COVID-19 patients received prophylactic or therapeutic anticoagulation, but additional data on exact dose for each patient was not available. Since we treated the primary exposure as a binary variable (i.e., prophylactic vs therapeutic anticoagulation), the lack of dosage data did not affect the results or conclusions reported in this study. The results of the present study identify a time-dependent association between survival and anticoagulant treatment regimen among severe COVID-19 patients in the ICU, which might provide useful guidance for future prospective studies that can consider further drug subtype analysis, such as new oral anticoagulants (NOACs) vs coumadin vs heparin effects on COVID-19 outcomes. As with any retrospective study, there could be unintentional patient selection bias, and therefore, randomized controlled trials are needed to assess causal relationships between various AC treatment regimens and COVID-19-related survival and long-term outcomes. Also inherent to cohort studies is the potential for unintentional confounding variables, such as group differences in age, although we attempted to control for this in covariate adjusted analysis.”

4. Indeed, in the therapeutic anticoagulation group, survival was linked to lower D-dimer, suggesting that some coagulation effect was present (Table 4).

We have only a Table describing survival parameters in the therapeutic dose group. Why you do not report a similar Table for the group without therapeutic anticoagulation?

Response: Thank you for providing us the opportunity to clarify this point. Table 4 represents a follow-up analysis based on our primary discovery that therapeutic dose users have increased risk of death after 3 or more weeks of ICU admission compared to prophylactic dose users, adjusted for age, clinical status, D-dimer and comorbidities. The goal of Table 4 is to better understand what factors were associated with poor survival in this particular group of interest. Reporting survival parameters for prophylactic users with prolonged ICU stay is of less interest in our opinion, because this group has favorable survival. We amended the text on page 11 to clarify this point:

“To better understand what factors were associated with poor survival among therapeutic anticoagulant users with 3 or more weeks of ICU admission, we conducted follow-up analysis comparing survivors (n=35) and non-survivors (n=32) in this group of interest (Table 4). Non-survivors were characterized by significantly older age (p=0.008), higher prevalence of coronary artery disease (p=0.009), lower lymphocyte count (p=0.007), and higher incidence of myocardial infarction (p=0.005). The majority (87%) of non-survivors were male (p=0.08).”

Thank you for your thoughtful comments.  

Comments to the Editor. We made the following additional revisions:

1. We made the title more specific to better reflect the study conducted.

a. Old title: Dose and duration effects of anticoagulant treatment among critically ill COVID-19 patients

b. New title: Prophylactic versus therapeutic dose anticoagulation effects on survival among critically ill patients with COVID-19

2. Minor text clarifications in the abstract: 

Introduction Although patients with severe COVID-19 are known to be at high risk of developing thrombotic events, the effects of anticoagulation (AC) dose and duration on in-hospital mortality in critically ill patients remain poorly understood and controversial. The goal of this study was to investigate survival of critically ill COVID-19 patients who received prophylactic or therapeutic dose AC and analyze the mortality rate with respect to detailed demographic and clinical characteristics.

Materials and Methods We conducted a retrospective, observational study of critically ill COVID-19 patients admitted to the ICU at Stony Brook University Hospital in New York who received either prophylactic (n=158) or therapeutic dose AC (n=153). Primary outcome was in-hospital death assessed by survival analysis and covariate-adjusted Cox proportional hazard model.

Results For the first 3 weeks of ICU stay, we observed similar survival curves for prophylactic and therapeutic AC groups. However, after 3 or more weeks of ICU stay, the therapeutic AC group, characterized by high incidence of acute kidney injury (AKI), had markedly higher death incidence rates with 8.6 deaths (95% CI=6.2-11.9 deaths) per 1,000 person-days and about 5 times higher risk of death (adj. HR=4.89, 95% CI=1.71-14.0, p=0.003) than the prophylactic group (2.4 deaths [95% CI=0.9–6.3 deaths] per 1,000 person-days). Among therapeutic AC users with prolonged ICU admission, non-survivors were characterized by older males with depressed lymphocyte counts and cardiovascular disease.

Conclusions Our findings raise the possibility that prolonged use of high dose AC, independent of thrombotic events or clinical background, might be associated with higher risk of in-hospital mortality. Moreover, AKI, age, lymphocyte count, and cardiovascular disease may represent important risk factors that could help identify at-risk patients who require long-term hospitalization with therapeutic dose AC treatment.

---

## [Decision Letter · Decision Letter 2]

6 Jan 2022

Prophylactic versus therapeutic dose anticoagulation effects on survival among critically ill patients with COVID-19

PONE-D-21-15647R2

Dear Dr. Duong,

We’re pleased to inform you that your manuscript has been judged scientifically suitable for publication and will be formally accepted for publication once it meets all outstanding technical requirements.

**The re-revised manuscript is definitely improved. The authors have properly addressed all the remaining comments of the reviewers.**

Kind regards,

Giuseppe Remuzzi

Academic Editor

PLOS ONE

Additional Editor Comments (optional):

Reviewers' comments:

Reviewer's Responses to Questions

**Comments to the Author**

1. If the authors have adequately addressed your comments raised in a previous round of review and you feel that this manuscript is now acceptable for publication, you may indicate that here to bypass the “Comments to the Author” section, enter your conflict of interest statement in the “Confidential to Editor” section, and submit your "Accept" recommendation.

Reviewer #1: All comments have been addressed

2. Is the manuscript technically sound, and do the data support the conclusions?

Reviewer #1: Yes

3. Has the statistical analysis been performed appropriately and rigorously? 

Reviewer #1: Yes

4. Have the authors made all data underlying the findings in their manuscript fully available?

Reviewer #1: Yes

5. Is the manuscript presented in an intelligible fashion and written in standard English?

Reviewer #1: Yes

6. Review Comments to the Author

Reviewer #1: All points have been addressed

I have nothing additional to add

7. PLOS authors have the option to publish the peer review history of their article (what does this mean?). If published, this will include your full peer review and any attached files.

Reviewer #1: **Yes: **JECKO THACHIL

---

## [Editor Report · Acceptance letter]

10 Jan 2022

PONE-D-21-15647R2 

Prophylactic versus therapeutic dose anticoagulation effects on survival among critically ill patients with COVID-19 

Dear Dr. Duong:

I'm pleased to inform you that your manuscript has been deemed suitable for publication in PLOS ONE. Congratulations! Your manuscript is now with our production department. 

Kind regards, 

on behalf of

Prof. Giuseppe Remuzzi 

Academic Editor

PLOS ONE